# Knowledge, Attitudes, Beliefs, and Practices Regarding Dengue in La Réunion Island, France

**DOI:** 10.3390/ijerph19074390

**Published:** 2022-04-06

**Authors:** Florence Lamaurt, Olga De Santis, Julie Ramis, Cédric Schultz, Ana Rivadeneyra, Mathias Waelli, Antoine Flahault

**Affiliations:** 1Institut de Santé Publique, d’Epidémiologie et de Développement (ISPED), Université de Bordeaux, 33000 Bordeaux, France; 2Faculty of Medicine, Institute of Global Health, University of Geneva, 1202 Geneva, Switzerland; simon.waelli@unige.ch (M.W.); antoine.flahault@unige.ch (A.F.); 3Service de Santé Publique et Soutien à la Recherche, Inserm CIC1410, CHU La Réunion, 97410 Saint-Pierre, France; 4Processus Infectieux en Milieu Insulaire Tropical, Université de La Réunion, 97490 Sainte-Clotilde, France; julie.ramis@gmail.com; 5Délégation à la Recherche Clinique et à l’Innovation, CHU La Réunion, 97410 Saint-Pierre, France; cedric.schultz@gmail.com; 6Bordeaux Population Health Inserm U1219, Université de Bordeaux, 33000 Bordeaux, France; ana.rivadeneyra@u-bordeaux.fr

**Keywords:** KABP, dengue, La Reunion, mixed-methods study

## Abstract

Since 2017, La Réunion island has been facing a major epidemic of dengue. Despite actions carried out by the anti-vector control department, public authorities have failed to contain this epidemic. As individual involvement is key to success in vector control, we carried out a mixed-methods study on population knowledge, attitudes, beliefs, and practices (KABP) regarding dengue infection risk in La Réunion. The study combined quantitative data collected through a questionnaire administered to a representative sample of 622 people to assess the use of protective measures and the perception of severity and risk of dengue, and a sample of 336 people to assess the level of knowledge and concern about dengue, as well as qualitative data collected through semi-structured interviews among 11 individuals who had previously completed the questionnaire. The study results show that 63% of the surveyed population had a good level of knowledge associated with age, education, and socio-professional category variables—78% considered dengue to be a serious threat, and concern was estimated at 6/10, while 71% were likely to use protective measures. The interviews revealed contradictory behaviors in the implementation of recommended actions, in conflict with personal beliefs regarding respect of human body and nature. The study also revealed a loss of confidence in public authorities.

## 1. Introduction

Dengue is the most prevalent arboviral disease in the world. The virus infects up to 390 million people living in tropical and sub-tropical areas [1]. The incidence of dengue has increased by 30-fold in the last 50 years due to global warming, increased urbanization, population growth, and the development of international travel [2].

The dengue virus is transmitted to humans by two species of *Aedes* mosquitoes: *Aedes aegypti* and *Aedes albopictus*. *A. aegypti* is the principal vector of transmission worldwide, while *A.albopictus* settles where *A. aegypti* is not very present, like in Indian Ocean islands [3].

The first documented outbreak of dengue in La Réunion took place in 1977–1978. The estimated attack rate was 30%. Between 2004 and 2017, only sporadic autochthonous cases were reported [4]. The most significant outbreak reached 200 cases in 2016. In March 2018, a new dengue epidemic broke out in the island and local authorities launched level 3 of the ORSEC plan (Organisation de la Réponse de Sécurité Civile), which is a program for organizing assistance at a departmental level in the event of a disaster. It allows for the rapid and effective implementation of all necessary resources under the authority of the prefect. This plan is aimed at efficiently organizing the response to limit the spread of the virus.

Vector control measures have existed in La Réunion since 1914, for the prevention of malaria, which was eradicated at the end of the 1970s. After a major outbreak of chikungunya in 2005, which infected 38% of the population, a specific vector control system was set up [5]. However, cases of dengue infection are currently reported from almost all municipalities, and three serotypes (DEN-1, DEN-2, and DEN-3) co-circulate on the island. In 2021, 29,222 confirmed cases and 19 deaths directly linked to dengue were reported [6].

To the best of our knowledge, no KABP study has been carried out on dengue in La Réunion. In 2008, one study focused on social, environmental, and behavioral factors during the outbreak of chikungunya [7]. One KABP study about dengue fever was set in Martinique (another French island) in 2008 [8]. Only a few studies have looked at the level of knowledge of the population and their perception of the disease [9,10,11]. Most were interested in the level of knowledge of schoolchildren or students, in order to improve school prevention programs [12,13,14,15].

According to the World Health Organization (WHO), a communities’ involvement is key to the success of vector control [16]. In addition, many studies have shown the importance of taking into account local knowledge and representations of vector-borne diseases when developing vector control strategies [8,17,18]. The aim of this study is to describe the knowledge, attitudes, beliefs, and practices (KABP) related to dengue transmission and prevention among the population living on La Réunion Island and to understand the impact of reported KABP on dengue’s shift to endemicity.

## 2. Materials and Methods

### 2.1. Study Sites and Population

This study was conducted as a part of a larger epidemiological research work, the DEMARE study, an observational cross-sectional epidemiological study conducted in La Réunion and Madagascar in 2019–2020. The main objective of the DEMARE study was to estimate the prevalence of dengue infections of all clinical forms, including asymptomatic cases in the community, according to a clustered geographical design based on dengue index cases.

Our study was carried out in 19 geographical clusters following the field design of the DEMARE study. Clusters were defined by a 200-m radius around a dengue index case. These clusters were mainly located in the west coast of La Réunion, where the epidemic was still raging during the study recruitment period. Participants were first contacted at their homes following a door to door recruiting schedule in order to arrange an inclusion appointment. This meeting allowed for collecting informed consent, administering the study questionnaire, and contacting participants who subsequently accepted being interviewed.

### 2.2. Study Design

This mixed-methods study combined quantitative and qualitative data collection. First, quantitative data were obtained from a questionnaire including socio-demographic variables, perception of danger related to dengue, and use of protective measures against mosquito bites. The calculation of the sample size was based on a known probability of good level of knowledge about dengue rate, estimated at 80% [8], a confidence level of 1.96 (95%), and an error of 5%. The resulting sample size was of 246 participants.

The DEMARE study included 622 participants. The KABP study was nested in the DEMARE study and questionnaires were administered to 336 participants. However, as part of the questions of the KABP study were asked in the DEMARE case report form, for some variables, data for 622 participants were available. Data were collected from 28 October 2019 to 27 August 2020. Data analyses were carried out using the statistical package R, version 3.4.4.

Second, qualitative data were obtained from semi-structured interviews using an interview guide following five main dimensions of disease representation: identity, temporality, causes, consequences, and controllability (Leventhal, 1980). A convenience sample was selected based on the principle of maximum variation on a number of variables: gender, level of education, socio-professional category, type of dwelling, place of birth, and data from the quantitative survey questionnaires (self-reported use of protective measures, feeling frequently being bitten or not by mosquitoes). The sample size was defined by data saturation when the interviews did not provide any new information.

A convenience sample of 11 interviewees was composed according to their age, sex, level of education, protective measures declared in the questionnaires, and housing conditions. Interviews were conducted at the participants’ home, in a calm environment and according to their availability. Each interview was recorded and completely transcribed. A thematic analysis of the collected data was carried out by two investigators using an individual open coding, and then confronting analyses and creating a new thematic.

## 3. Results

### 3.1. Results of the Quantitative Study

#### 3.1.1. Level of Knowledge about Dengue

Knowledge about dengue was rated by six questions concerning disease classification, vector, transmission, severity, symptoms, and immunology. The questions could be answered by “yes”, “no”, or “I do not know”. Here are the questions with the correct answer rate in brackets (CAR): (1) Is dengue a disease caused by bacteria? (CAR = 53%); (2) Can all mosquitoes transmit dengue? (CAR = 82%); (3) Can dengue be transmitted through saliva? (CAR = 75%); (4) Can you get sick without having any symptoms? (CAR = 79%); (5) Can dengue be a deadly disease? (CAR = 97%); and (6) Can you get dengue more than once? (CAR = 86%). The individual CAR oscillated between 17% and 100%. The mean of individual CAR was 79%.

The level of knowledge was defined as “poor” for CAR ≤ 25%, “insufficient” for CAR > 25 ≤ 50%, “medium” for CAR > 50 ≤ 70%, and “good” for CAR > 70%. Figure 1 shows the repartition of level of knowledge about dengue.

The cross-analysis between the level of knowledge and the other study variables is detailed in Table 1. Advanced age (*p* < 0.05), none or primary education (*p* < 0.05), and professional inactivity (unemployed, retired, or disabled) (*p* < 0.05) are associated with a poor level of knowledge. These variables were adjusted in a logistic multivariate model that showed interactions between age and professional inactivity, and between age and a poor level of education. These determinants were linked and concerned the same subset of the population.

#### 3.1.2. Perception of Danger Related to Dengue

The perception of danger was rated by two questions: “What about the seriousness of having dengue fever?” and “Do you think you are at risk of contracting dengue in the next five years?”. Figure 2 and Figure 3 show the distribution of answers.

The cross-analysis between the perception of severity and other studies variables is detailed in Table 2. Gender, age, level of education, and socioprofessional categories were associated with the perception of severity of dengue fever. There was no association between the knowledge about dengue and the perception of severity.

The cross-analysis between the perception of risk and other studied variables is detailed in Table 3. School children or students were the less concerned with the risk of contracting dengue, contrary to state employees who were the most concerned. People who had already experienced dengue-like symptoms were more susceptible to perceive a high risk of contracting the disease. Unsurprisingly, participants who declared never having been bitten by mosquitos were more susceptible to perceive a low risk of contracting dengue. The cross-analysis also highlighted an association between the level of knowledge about dengue and the perception of risk (*p* = 0.01).

Participants were also asked to express their level of personal concern about five other health risks by giving them a score from 0 to 10 in the same way, namely: road accidents, seasonal flu, cyclones, diabetes, and chemicals in food. This technique made it possible to situate the concern about dengue in a broader context of multiple health risks to which the people of Reunion are exposed. The results are presented in Figure 4. The median of the dengue-related concern was of 6/10 (range 0–10). Dengue was less worrying than road accidents (8/10), use of pesticides (7/10), and diabetes (7/10), but more worrying than cyclone (5/10) and seasonal flu (5/10).

#### 3.1.3. Use of protective Measures against Mosquito Bites

Here, 71% of participants reported using protective measures against mosquito bites, where 59% used it “sometimes” and 35% “daily”. Only 6% reported using protective measures “very rarely”.

Reported protectives measures are shown in Figure 5. The most frequent answer was “other measures of protection”, including essential oils (lemongrass and geranium), UV lamps, mosquito nets on windows, insecticides use in gardens, fishponds, and smoke (traditional method of burning herbs and leaves, while cleaning gardens to keep mosquitoes away).

Participants were also asked about their perception of the efficiency of different protective, individual, and collective measures, by rating them from 0 (not efficient) to 10 (extremely efficient). The results are presented in Figure 6. The measure considered the most efficient was the elimination of breeding sites in gardens or courtyards (with a median of 9/10 and a low dispersion). The other measures rated by the participants included, in decreasing order, maintenance of gullies by municipalities, mosquito nets, skin repellent lotions, mosquito control program, and indoor repellents. These results have to be considered with caution, as answering this question was complex for participants for two reasons: a moral dilemma between efficiency and environmental harmfulness, and between theoretical and real efficiency.

The cross-analysis between the use of protective measures and other studied variables is detailed in Table 4. This analysis shows that respondents’ sex, the level of education, the professional activity, and having the idea of being bitten by mosquitos would significantly influence the use of protection measures. Indeed, the variables associated with the use of protection measures were “women”, “high level of education”, and “impression to be frequently bitten by mosquitos”. On the contrary, variables associated with a decreased use of protection measures included “schoolchildren or students” and “impression to be never bitten by mosquitos”. The perception of danger did not significantly influence the use of protective measures. Moreover, the level of knowledge about dengue did not significantly influence the use of protective measures.

### 3.2. Results of the Qualitative Study

The sample was composed of five men and six women, between 30 and 70 years old, with an average age of 55. One was unemployed, five worked in public or private firms, and five were retired. Five were born in Metropolitan France, one living in La Réunion for 4 years and four for more than 5 years. Six were born in La Réunion. Four of them had already contracted dengue, biologically confirmed.

#### 3.2.1. Knowledge and Beliefs about Dengue

The overall knowledge about dengue was good among interviewees. They compared dengue with chikungunya or seasonal flu, because of their similar symptoms. The 2005–2006 chikungunya epidemic was a vivid memory, whether people got sick or not. The symptomatology of dengue was known by all interviewees, but descriptions varied from one participant to another. They all insisted on the suddenness and the long duration of the disease.

Interviews showed different interpretations of the word “dengue”. Firstly, for many interviewees, dengue was a vector-borne disease transmitted by the “tiger mosquito”. However, for elderly people, the word dengue was designated an important flu syndrome not related to mosquitoes. They estimated that dengue was an airborne disease, and that the “tiger mosquito” had disappeared from the island following the chikungunya epidemic.

Interviewees also showed a lack of knowledge about immunology and the different serotypes. They often did not know if dengue could be contracted more than once. Some of them thought that one serotype was more dangerous than another, and had no idea about secondary infection and the related risk of increased severity.

#### 3.2.2. Perception of Danger

The concept of risk was present among all interviewees, and it was linked to the geographical and social proximity of the epidemic. As a result of its presence on the island for several years, dengue appeared to be a common disease: people got used to it because they, or their relatives, had contracted it. Most of interviewees expressed concern for others: the elderly, people with chronic pathologies, or young children.

The severity and mortality of dengue were also discussed with interviewees. They felt concerned as the disease seemed to get more severe from year to year, and often mentioned number of deaths and possible comorbidities, and frequently compared the number of deaths to that associated to chikungunya or COVID-19 (the COVID-19 crisis was widely covered by the media over the study period). Interviewees often considered dengue as a less lethal disease than chikungunya or COVID-19, and pointed out a lack of information about its severity. Sometimes, they assumed dengue victims presented comorbidities.

#### 3.2.3. Perception of Local Ecology

Interviewees who had grown up in La Réunion shared the idea that the island was naturally protected due its remoteness. Nevertheless, they also thought that La Réunion had been subjected to critical changes in recent years, including growing urbanization and increasing international travel and tourism, that disrupted its natural protection. Within this context, dengue was considered as an imported disease.

All interviewees thought that mosquitoes were part of La Réunion historical fauna and for many of them, they were not responsible for any disease in the past (although malaria was once endemic before it was eradicated in the late 1970s). Finally, they were well informed of risks factors for dengue transmission. Having been exposed to malaria and chikungunya epidemics, they were aware of the risks and knew how to protect themselves against vector-borne diseases.

#### 3.2.4. Responsibility for Prevention

Three levels of responsibility in the fight against dengue were identified: individual responsibility, responsibility of municipalities, and responsibility of the Health Regional Agency (Agence Régionale de Santé, ARS).

Regarding individual responsibility, prevention campaigns seemed to have achieved their objective: all interviewees mentioned cleaning their garden and removing stagnant water tanks. Field observations confirmed this practice, even among participants who refuted the role of mosquitoes in disease transmission. Most of the interviewees mentioned the use of individual prevention measures such as body sprays, spirals, long clothes, mosquito repellent plugs, air blowers, air conditioning, and essential oils. These protecting measures were used to protect oneself or to protect others, as a civic duty, especially when one was contagious. However, such practices were only occasional, when mosquitoes were visible or during high-risk periods.

The responsibility of municipalities was involved in the various means deployed to clean up public areas and to stimulate individual responsibility. Environmental pollution was identified as a risk factor leading to mosquito proliferation. Garbage, car wrecks, and any other material found on waysides are reservoirs for mosquitoes on the island as soon as it rains. Acting on fly-tipping was repeatedly mentioned during the interviews as an important lever for action in vector control. Verbalizing polluters seemed to be a necessary solution in response to the lack of civic mindedness, and the state of cleanliness of cities was criticized. While interviewees felt their individual responsibility was constantly challenged through prevention campaigns, they considered that public areas were not as clean as their own places.

The role of the ARS was frequently mentioned in relation to the vector control strategy based on door-to-door or night trucks campaigns. The interviewees considered that there is a lack of clarity regarding the organization of these actions. Some interviewees mentioned that they were still waiting for the ARS to pass by their homes, and one explicitly declared that it passed very seldomly. Others felt that the employed methods were inconsistent: spraying the top of piles of leaves, next to piles of grass, or not treating ponds. The treatment of ravines was questioned by most of the interviewees. According to them, while the prophylaxis service acting against malaria was used to disinfect the ravines, and this prevention measure was no longer carried out. Nobody knew for sure why this prophylaxis had stopped, and some interviewees evoked potential reasons such as banning related to ecological risks (ocean pollution) or difficulty of access. All participants considered ravines as the largest mosquito reservoir and estimated that they should be a priority target for the ARS, which should act where individuals and municipalities cannot gain access.

#### 3.2.5. Individual Beliefs: A Barrier to Prevention?

Our analysis reveals some beliefs that could impact dengue prevention behaviors. Individual or collective protection was considered necessary, in accordance with individual beliefs and two main imperatives: respect for one’s body and respect of nature. The use of anti-mosquito sprays was occasional because of limited knowledge about chemicals and related long-term effects. Interviewees were more willing to use individual protection measures if they were natural. Those who chose to protect themselves did so in accordance with their own health-related values.

The role of insecticides in the mosquito control measures carried out by the ARS was strongly contested in the interviews. Few people were totally opposed to them, but they had many questions about the dangerousness and the effectiveness of the products spread. The impact of insecticides on other animals was highlighted: wasps, lizards, birds, chameleons, bees, fish, spiders, etc. A vicious circle was pointed out: the disappearance of the species regulating the populations of mosquitoes. Not totally opposed to insecticides, several interviewees were eager to change the products currently used by the ARS, on the one hand, to achieve more efficacy, because mosquitoes had developed resistance to the product used for years and, on the other hand, to use insecticides specifically targeting against mosquitoes and not killing other harmless species.

#### 3.2.6. Motivation and Confidence in Public Authorities

Public authorities were strongly criticized by almost all of the interviewees, evidencing a significant loss of confidence in public authorities. They felt that many preventive actions were conducted without any results and the lack of feedback was clearly condemned.

In addition, prevention campaigns were described as oppressive by some interviewees. They expressed the feeling of being constantly blamed by public authorities, while many other actions could be carried out by the municipalities or the regional authorities.

#### 3.2.7. Vision of the Future

Opinions were divided among interviewees. Those more fatalistic thought dengue would subsist because mosquitoes would not be extinct in La Réunion and preventive actions were thus done in vain. Those more optimistic believed actions could be conducted to control the spread of the epidemic. Nevertheless, some of them thought the existing vector control strategy and the resulting actions were not suitable and they had some proposals to define new strategies. Others believed that advances in biology and medicine research would help find a solution.

## 4. Discussion

The main findings of our study concern the high level of knowledge about dengue among the population, the link between the perception of risk and the adoption of individual protection measures, and the existence of constraints to social mobilization.

More than 97% of participants were informed about dengue. Two thirds of the participants had a good level of knowledge about dengue; a similar result was found in the West Indies [8]. The level of knowledge was related to the age, the education level, and the socio-professional category, which was consistent with the literature on the determinants of health knowledge [19]. We did not find a link between the level of knowledge and antecedents of dengue, which has been highlighted in other studies [11].

Knowledge on dengue vectors and its mode of transmission was also well integrated: 82% of interviews knew that not all mosquito species can transmit dengue and 75% knew that dengue was not transmitted through saliva. Some secular beliefs persisted, especially linked to the different meanings of the word “dengue” in La Réunion. Indeed, the word dengue represented “mosquito bite”, but also a set of symptoms similar to dengue-like symptoms or flu-like symptoms. Seasonal influenza had long been referred by doctors as dengue, a synonym widely accepted on the island [5]. A previous study already highlighted this point [20]. Dengue was associated with vector transmission only in 2012. The actual vector control service was created after the chikungunya epidemic and the first prevention campaign against dengue was conducted in 2012 [5,20].

The existence of several serotypes of dengue, as well as the concept of secondary dengue was very poorly known and understood. The result to the question “Can you have dengue more than once?” was certainly overestimated, because the recruitment in the field allowed for a first exchange with the interviewees and answering some of preliminary questions. Many of them believed that there were several types of dengue, one of which was more serious than the others. Providing information about the existence of four different dengue serotypes would allow individuals to be more aware of the risk of new infection and be more involved in the application of prevention measures, particularly in a context of potential low recourse to healthcare. In fact, dengue diagnosis has not been confirmed in the laboratory for more than a third of people who declared having contracted the disease. However, these findings must be taken with caution due to the COVID-19 pandemic, which might have introduced a bias. Indeed, the lock-down might have discouraged some people to seek laboratory tests.

Dengue seemed to be considered as a relatively serious disease for almost 80% of the surveyed population. More than 75% considered that they would likely contract dengue in the coming five years. Dengue was a moderate source of health concern (median score of 6/10). On a health risk scale, dengue ranked fourth after road accidents, use of pesticides, and diabetes, but ahead of cyclones or flu. This paradox was highlighted in a similar study carried out in 2008 in Martinique [8]. Our qualitative study showed that dengue severity was often compared to chikungunya, which killed around 150 people in 2005–2006. The death rate of dengue was generally not well known, and the existence of associated comorbidities was still suspected. Interviewees felt more concern for others, especially for the elderly and children, than for themselves. Cross-analysis showed associations between the perception of dengue risk and the knowledge on the vector, as shown in other studies [10].

More than 70% of the surveyed population used a protective measure against mosquito bites. This represents a higher frequency compared to various studies carried out on the same subject in La Réunion [8,21]. Skin repellents were relatively barely used, while they are recommended as one of the most effective methods (BEH, 2012). The use of mosquito coils (spirales), a less efficient protection (BEH, 2012), was significant even though it decreased compared to 2014. However, the perceived effectiveness of these devices (sprays and mosquito coils) slightly increased in La Réunion since 2014 [21]. Protective measures were not used daily, but on an occasional basis.

The level of individual protection was not significantly linked to the perception of severity or risk. This surprising observation was already described in a qualitative study carried out during the chikungunya epidemic in La Réunion [7]. This result contradicts that of another study underlining that these two variables are critical in the adoption of protective measures (Slovic, 1999). Our results showed no significant link between the experience of dengue and the attitude towards dengue control, which contradicts other studies [10]. In addition, the use of protective measures did not reflect the level of individual knowledge, while many other studies stressed the importance of knowledge to the perception of risk and the implementation of protective measures [11,22]. The reasons people did or did not use individual protection measures can only be understood through qualitative studies, which should be used more often. The use of an alternative protection method was frequently cited. Its effectiveness was not scientifically proven, but acceptability was considered more important. Scoring the efficiency of the listed protection measures was difficult for the study population. Protective measures were a source of moral dilemma when balancing the benefits and health risks. The need for a benefit–risk approach taking into account safety and efficiency of protective measures has been mentioned in other studies [5,21].

After the chikungunya epidemic, the anticipation of epidemics became a central point of health policies. To date, the lack of confidence in the health authorities seems to be a barrier to social mobilization. The place of insecticides in vector control and its consequences on the environment in the short or long term take an important place in the public debate. Beyond questioning the dangerousness of chemical products, which is mainly due to a lack of clear and easily accessible information or convincing studies, it is above all a lack of organization, and results in the general fight against dengue that is pointed out by the participants of the study.

The qualitative analysis defined three levels of responsibility in the fight against mosquitoes: individuals, municipalities, and the ARS. Prevention campaigns are still mainly focused on populations only, as underlined by the ANSES (Agence nationale de sécurité sanitaire de l’alimentation, de l’environnement et du travail) report on the vector control strategy in 2018 [5], and as also pointed out by several interviewees. By linking the spread of mosquitoes to human practices, authorities held the population responsible for mosquito proliferation, and thereby for dengue proliferation. This was frequently used to hide the difficulties of the public authorities in dealing with the epidemic [20]. Convinced of doing their best at home, interviewees naturally shifted responsibility to public authorities.

Strongly committed from the start of the epidemic in 2017, municipalities initially stepped up their actions of waste collection, road cleaning, and elimination of fly-tipping, in particular by hiring jobs. As the years passed by, it appeared to lose some strength [5]. A remobilization of the municipal level seems to be essential for a collective vector control strategy and to re-engage the community.

Regarding the action of the ARS, this study showed a perceived lack of organization and training of workers for peri-domiciliary mosquito control. There were doubts about the effectiveness of the strategy employed, in particular due to the total lack of information on the results obtained. The adequacy between the means implemented regarding the objectives and the feedback on the effectiveness of the measures was regularly challenged. Effectiveness studies on the impact of the measures taken would be necessary to regain the confidence of the population. The ultra-vertical system and the complexity of the administrative organization in the governing bodies are often criticized [5]. In 2005–2006, the low-risk perception of chikungunya in La Réunion resulted in poor adherence to the vector control strategy that was put in place to tackle the epidemic [7]. Today, faced with a higher perception of risk and a strong perception of danger, communication campaigns should be part of a more horizontal approach by opening a dialogue with the population and by moving away from standardized messages sometimes badly perceived.

Finally, the results of the interviews suggest the emergence of a “dengue culture” linked to the proximity of the vector and the disease. This observation was also made in Martinique, where the epidemic context was similar [8]. It was found from our interviews that the population of La Réunion was used to living with mosquitoes. The elderly said mosquitoes have always been there, unlike diseases. Even the youngest or the most recently arrived on the island said that living with mosquitoes in La Réunion was usual. Mosquitoes occupy an important place in the local ecology. Directing control action solely on the eradication of the vector seemed to raise doubts on the efficiency.

### Strengths and Limits

This study is an original attempt to identify the level of knowledge, beliefs, attitudes, and practices of the population living in La Réunion considering the dengue epidemic that began in 2017. It was carried out on a statistically representative sample of 622 people (with some questions addressed to a sub-group of 336 people), well beyond what was recommended by the sample size calculation in the study protocol. The quantitative study was based on a protocol and an analysis plan defined upstream of the study (internal validity). Some variables used came from previous studies carried out in La Réunion or in other DROM (Département et Régions d’Outre-Mer) on dengue or other arboviruses, in order to allow for external validity of the study. The use of a qualitative approach has shown its relevance to complement the quantitative approach and to understand different representations of the disease.

The main limitation of the study is the short time allowed to implement it. Fieldwork remains a complex issue, especially in the context of the COVID-19 crisis. For the qualitative study, interviews could be conducted only after the lifting the lockdown (11 May 2020). The health protocol imposed a strict procedure for disinfecting the equipment of the DEMARE study (serology test) during which this study was conducted, between each participant as well as the fitting out of a vehicle for the needs of this procedure.

The possible biases of our study are the following:(i)Quantitative recruitment bias: field recruitment took place during the week, in normal working hours. Indeed, the sample showed a high proportion of retired people, who were more available at this time slot.(ii)Information biases: with certain variables being declarative, it is relevant to think that estimates are over or underestimated. For example, the frequency of use of protective measures may have been overestimated, as participants were able to respond by compliance with the obligations conveyed in prevention campaigns.(iii)Confounding factors: the variable “type of environment” was not used to conclude on the results of the study. Indeed, this data cannot be used as such and should be subjected to additional analysis relating to the social level with additional data that were not available.

## 5. Conclusions

This study indicated that the population of La Réunion is very well informed about the dengue virus, its severity, and the current individual risk on the island. This population is used to living with mosquitoes, and it seems utopian for them to think of the island without mosquitoes. This leads to poor adherence to vector control programs. The concerns about toxic products used for individual or collective protection measures is real and must be taken into account by the public authorities to allow for effective social mobilization. Finally, interviewees asked for more feedback and transparency on the prevention actions conducted by health authorities.

## Figures and Tables

**Figure 1 ijerph-19-04390-f001:**
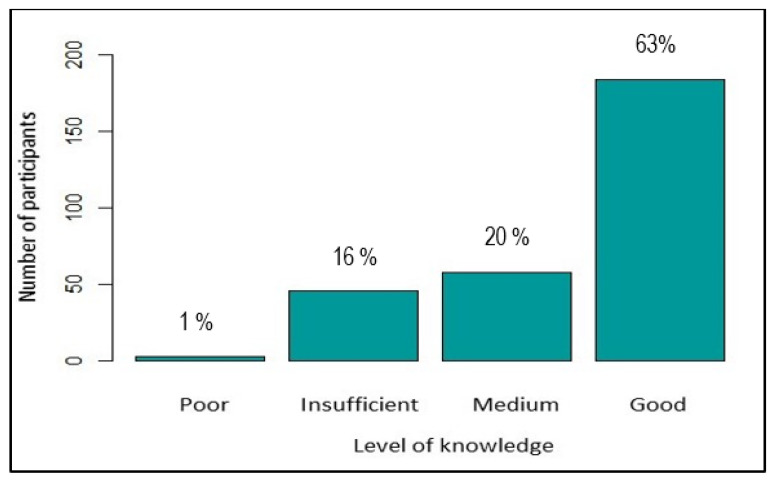
Level of knowledge about dengue.

**Figure 2 ijerph-19-04390-f002:**
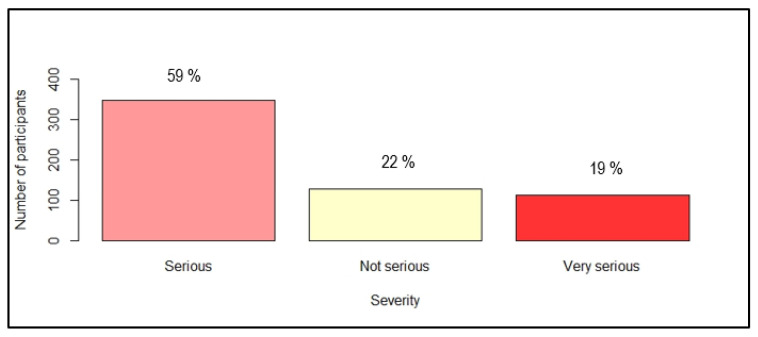
Distribution of participants according to their perception of severity of dengue (*n* = 587).

**Figure 3 ijerph-19-04390-f003:**
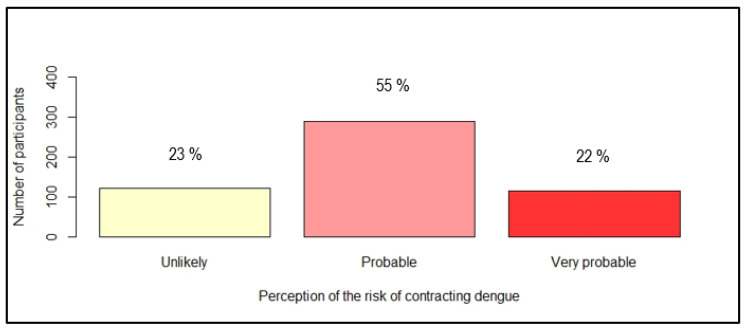
Distribution of participants according to their perception of the risk of contracting dengue in the coming five years (*n* = 525).

**Figure 4 ijerph-19-04390-f004:**
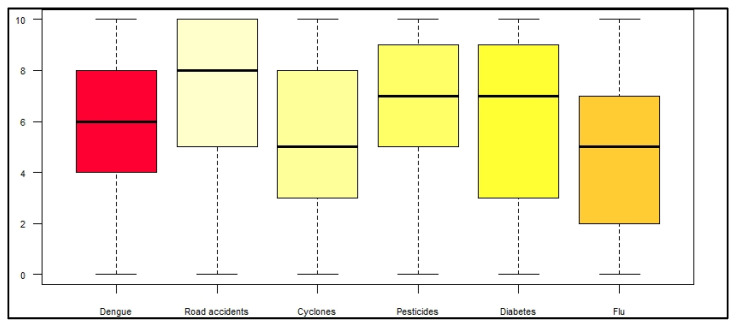
Degree of concern about dengue regarding other potential risks.

**Figure 5 ijerph-19-04390-f005:**
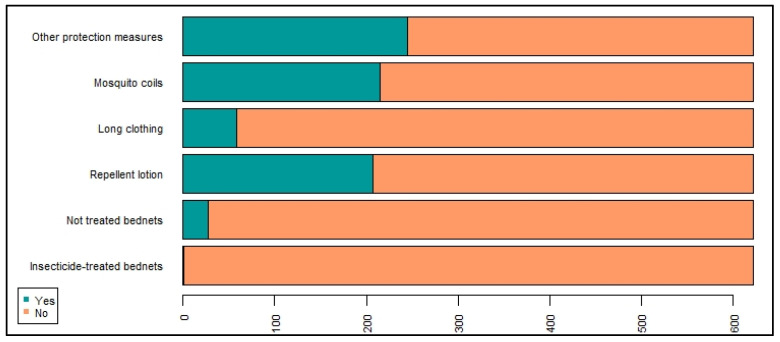
Repartition of the different protective measures reported by study participants.

**Figure 6 ijerph-19-04390-f006:**
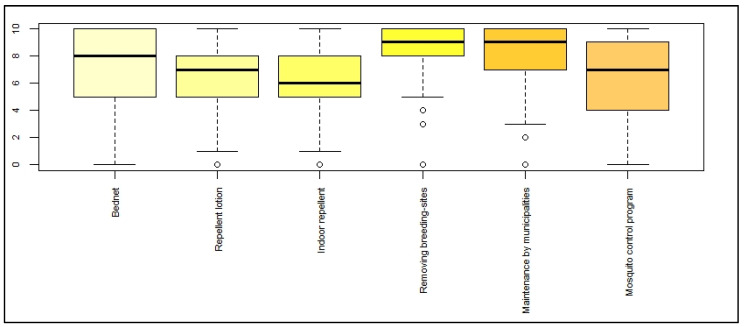
Perception of efficiency of different protective measures against mosquito bites.

**Table 1 ijerph-19-04390-t001:** Cross analysis between demographic characteristics or the experience of a dengue-like symptoms and the level of knowledge.

LEVEL OF KNOWLEDGE							
	Poor or insufficient		Medium		Good		*p*
	n	%	n	%	n	%	
**GENDER**							0.2
Female	33	18	29	16	118	66	
Male	16	14	29	26	66	60	
**AGE**							<0.005
Mean	61		52		49		
IC95%	57—66		48—57		47—51		
**LEVEL OF EDUCATION**							<0.005
None or primary	24	46	14	26	15	28	
Secondary 1	20	21	25	27	49	52	
Secondary 2	3	6	12	23	37	71	
Bac + 2	1	4	3	11	22	85	
Bac > +2	1	2	4	6	61	92	
**SOCIOPROFESSIONNAL CATEGORY**							<0.005
Schoolchild or student	0	0	3	18	14	82	
Housewife	5	17	7	24	17	59	
State employee	4	10	4	10	32	80	
Private employee	3	7	6	13	37	80	
Independant	1	4	7	25	20	71	
Unemployed	12	24	17	35	20	41	
Retired	24	30	14	18	41	52	
**USE OF PROTECTIVE MEASURE**							0.5
Yes	32	15	44	21	134	64	
No	17	21	14	17	50	62	

**Table 2 ijerph-19-04390-t002:** Cross analysis between socio-demographic factors, previous experience of a dengue-like symptoms, level of knowledge about dengue, and perception of severity.

PERCEPTION OF SEVERITY							
	Not serious		Serious		Very serious		*p*
	*n*	%	*n*	%	*n*	%	
**GENDER**							0.01
Female	59	17	210	62	69	20	
Male	68	27	138	56	42	17	
**AGE**							0.01
Mean	46		49		50		
IC95%	42—49		47—51		57—54		
**LEVEL OF EDUCATION**							0.02
None or primary	15	15	68	67	19	19	
Secondary 1	32	18	112	62	36	20	
Secondary 2	23	18	76	60	27	21	
Bac + 2	14	27	29	58	9	17	
Bac > +2	42	34	61	50	20	16	
**SOCIOPROFESSIONNAL CATEGORY**							0.004
Schoolchild or student	16	21	51	67	9	12	
Housewife	10	17	38	64	11	19	
State employee	21	27	36	46	21	27	
Private employee	23	29	47	59	10	13	
Independant	20	37	21	39	13	24	
Unemployed	12	16	48	65	14	19	
Retired	23	15	101	65	31	20	
**ANTECEDENT OF DENGUE-LIKE SYNDROM**							0.6
Yes	51	24	126	59	38	18	
No	76	20	221	60	74	20	
**FEELING OF BEING BITTEN BY MOSQUITOES**							0.6
Often	49	22	130	58	47	21	
Occasionnaly	37	26	78	56	25	18	
Rarely	29	17	107	64	31	19	
Never	12	23	32	60	9	17	
**LEVEL OF KNOWLEDGE ABOUT DENGUE**							0.3
Poor or insufficient	32	27	65	55	22	18	
Medium	61	21	174	61	51	18	
Good	22	19	63	56	28	25	

**Table 3 ijerph-19-04390-t003:** Cross analysis between socio-demographic factors, previous experience of a dengue-like symptoms, level of knowledge about dengue, and perception of risk.

PERCEPTION OF RISK							
	Unlikely		Probable		Very probable		*p*
	*n*	%	*n*	%	*n*	%	
**GENDER**							0.2
Female	59	19	175	58	69	23	
Male	62	28	113	51	46	21	
**AGE**							0.4
Mean	42		49		46		
IC95%	43—51		47—51		43—49		
**LEVEL OF EDUCATION**							0.2
None or primary	21	27	38	49	19	24	
Secondary 1	35	23	93	60	26	17	
Secondary 2	29	25	65	57	21	18	
Bac + 2	10	20	29	59	10	20	
Bac > +2	24	19	62	50	39	31	
**SOCIOPROFESSIONNAL CATEGORY**							0.0006
Schoolchild or student	25	40	28	45	9	15	
Housewife	12	22	29	54	13	24	
State employee	10	13	41	53	27	35	
Private employee	11	14	46	61	19	25	
Independant	6	13	26	54	16	33	
Unemployed	16	25	37	57	12	18	
Retired	38	29	75	57	19	14	
**ANTECEDENT OF DENGUE-LIKE SYNDROM**							0.006
Yes	30	16	108	57	51	27	
No	90	27	181	54	64	19	
**FEELING OF BEING BITTEN BY MOSQUITOES**							0.0002
Often	35	17	109	53	60	30	
Occasionnaly	24	19	79	62	24	19	
Rarely	44	30	76	51	28	19	
Never	18	40	25	56	2	4	
**LEVEL OF KNOWLEDGE ABOUT DENGUE**							0.01
Poor or insufficient	9	25	20	56	7	19	
Medium	15	28	32	61	6	11	
Good	22	13	101	58	51	29	

**Table 4 ijerph-19-04390-t004:** Cross analysis between use of protective measures, socio-demographic characteristics, experience of a dengue-like symptoms, and perception of severity or risk.

	USE OF PROTECTIVE MEASURES AGAINST MOSQUITO’BITES				
	Yes		No		*p*
	*n*	%	*n*	%	
**GENDER**					0.003
Female	268	75	87	25	
Male	170	64	94	36	
**AGE**					0.0002
Mean	49		42		
IC95%	48—51		36—49		
**LEVEL OF EDUCATION**					0.05
None or primary	79	64	45	36	
Secondary 1	133	73	49	27	
Secondary 2	83	65	45	35	
Bac + 2	41	77	12	23	
Bac > +2	99	77	29	23	
**SOCIOPROFESSIONNAL CATEGORY**					0.001
Schoolchild or student	50	52	46	48	
Housewife	45	76	14	24	
State employee	64	79	17	21	
Private employee	59	73	22	27	
Independant	40	74	14	26	
Unemployed	52	68	25	32	
Retired	121	76	39	24	
**ANTECEDENT OF DENGUE-LIKE SYNDROM**					0.1
Yes	166	74	57	26	
No	271	68	125	32	
**FEELING OF BEING BITTEN BY MOSQUITOES**					0.004
Often	175	75	57	25	
Occasionnaly	113	75	38	25	
Rarely	118	67	58	33	
Never	31	53	27	47	
**PERCEPTION OF SEVERITY**					0.2
Not serious	83	65	44	35	
Serious	255	73	93	27	
Very serious	84	75	28	25	
**PERCEPTION OF RISK**					0.1
Unlikely	86	71	35	29	
Probable	206	68	83	32	
Very probable	93	73	22	27	
**LEVEL OF KNOWLEDGE ABOUT DENGUE**					0.4
Poor or insufficient	32	65	17	35	
Medium	44	76	14	24	
Good	134	73	50	27	

## Data Availability

The data presented in this study are available on request from the corresponding author. The data will be publicly available in a data repository (https://yareta.unige.ch/#/home, accessed on 31 March 2022) as soon as all the manuscripts referring to the DEMARE study will be ready for publication.

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
