# Peer review of "Knowledge, Attitudes, Beliefs, and Practices Regarding Dengue in La Réunion Island, France"

_ijerph, 2022, doi:10.3390/ijerph19074390_

Round 1

Reviewer 1 Report

Dengue is one of the most prevalent arboviral diseases worldwide. The virus infects millions of people living in tropical and sub-tropical areas.

But the manuscript needs some points to be improved.

First, the introduction section contains a minimal number of references, the authors should include recent references related to the research topic.

for example:

Knowledge, attitudes, and practices towards dengue prevention among primary school children with and without experience of previous dengue infection in southern Thailand.Published online 2021 Jun 7. doi: 10.1016/j.onehlt.2021.100275

 -Ferreira-de-Lima V.H., Lima-Camara T.N. Natural vertical transmission of dengue virus in Aedes aegypti and Aedes albopictus: a systematic review. Parasit. Vectors. 2018;11:1–8. doi: 10.1186/s13071-018-2643-9.

Lauer S.A., Sakrejda K., Ray E.L., Keegan L.T., Bi Q., Suangtho P. Prospective forecasts of annual dengue hemorrhagic fever incidence in Thailand, 2010–2014. Proc. Natl. Acad. Sci. 2018;115:E2175–E2182. doi: 10.1073/pnas.1714457115.

Yacoub S., Farrar J. 15 - Dengue. In: Farrar J., Hotez P.J., Junghanss T., Kang G., Lalloo D., White N.J., editors. Manson’s Tropical Infectious Diseases. 23th edition. W.B. Saunders; London: 2014. pp. 162–170. e162

Thomas S.J., Yoon I.-K. A review of Dengvaxia®: development to deployment. Hum. Vaccin. Immunother. 2019;15:2295–2314. doi: 10.1080/21645515.2019.1658503.

Suwanbamrung C., Kusol K., Tantraseneerate K., Promsupa S., Doungsin T., Thongchan S. Developing the participatory education program for dengue prevention and control in the primary school, southern region, Thailand. Health. 2015;7:1255–1267. doi: 10.4236/health.2015.710140.

and others references related to the topic of the study

Results section

the authors should include the distribution of participants’ responses to items in the dengue-related knowledge in their results.

Correlations between knowledge, attitude, and practice scores towards dengue prevention should be identified in the author's results.

The discussion section needs to be more focused on the obtained data. a lot of comparisons between the author's data and others were missed.

Reviewer 2 Report

Line 37 - "has increased by 30-fold" is a better alternative

Lines 45 - 47. This sentence is too long and confusing. Suggest breaking it to 2-3 short sentences

Line 48 - There is no explanation as to what ORSEC involves 

Lines 52 - 55 seemes to be irrelevant to this paper as it is not focussing on vector control measures for DENV (noting that the vectors for chikungunya and dengue are the same)

Perhaps the introduction would benefit from briefly mentioning the results of KA(B)P studies previously conducted in Reunion islad or elsewhere on dengue.

There is no sample size calculation for the quantitative survey and it is unclear on what basis a subgroup of 336 particpants were administered some "additional" questions.

For the qualitative component there is no justification as to how or why these 11 people were selected for interviews. There is also no indication if thematic saturation was observed within 11 interviews

Lines 117 - 119. The questionnaire needs to provided as an English translation. It is not clear what questions were asked, how they were graded and whether this questionnaire is adequate to define a "good" vs. "bad" knowledge of dengue

Line 132 - "Bad level of knowledge" is not a proper term. Suggest changing to "insufficient knowledge"

Table 1 - some words are in French - please change to English

Lines 133 - 135 - It is unclear what authors mean by poor education or professional inactivity

Line 145 - "“What do you think about the 145 severity of contracting dengue?” is a rather meaningless question unless the authors asked a different question in French and this is a poor translation of that question

Line 146 - "“What do you think about the risk of contracting 146 dengue in the coming five years?” - Again I do not see the relevance of asking this question. Is there an increased risk of contracting dengue in the next five years at Reunion? Can anyone guarntee that this risk will increase or decrease?

I find the entire section 3.1.2 poorly designed and therefore lacking scientifically useful information. For example on what basis can someone compare dengue to RTA, cyclones or pesticides? What is the justification for selecting these items to compare?

Line 208 - Avoid starting sentences with numbers

The entire results section is too long and there is unnecessary repetition of information. Most sections could have been written succicnctly

The discussion is poorly presented without a systematic comparison to previous literature. Previous studies are only haphazardly referred to without any meaningful structure. For example in 423, authors mention that two thirds of participants had a good knowledge of dengue similar to a study in West Indies. But why would they want to compare with West Indies? What is the similarity of people or circumstances in Reunion and West Indies?

Lines 441 - 443 - It is unclear to me how an average person knowing about dengue serotypes can help them in avoiding complications. Do you routinely do RT-PCR and serotype patients within the first 3 days of fever and tell the patient the infecting serotype so that if they have second infection with a different serotype they can look out for serious complications. Unless this is the setup, I do not see how serotype information can be useful at an individual patient level.

Overall this paper would be of limited interest to a global readership. There are some significant shortcomings in study design and presentation of data.  The quality of written English needs improvement.

Round 2

Reviewer 1 Report

The authors responded to all comments and I believe the manuscript has been sufficiently improved to warrant publication in IJERPH.

Author Response

Thank you very much for your comments on our manuscript. 

We try to improve our English but it's difficult when it's not our mother tongue. 

Thank you for your understanding. 

Kind regards,